# Trend Analysis of Average Frequency Using Toothbrushing per Day in South Korea: An Observational Study of the 2010 to 2018 KNHANES Data

**DOI:** 10.3390/ijerph18073522

**Published:** 2021-03-29

**Authors:** Yu-Rin Kim, Hyun-Kyung Kang

**Affiliations:** Department of Dental Hygiene, Silla University, 140 Baegyang-daero 700 Beon-gil, Sasang-gu, Busan 46958, Korea; dbfls1712@hanmail.net

**Keywords:** oral health, South Korea, toothbrushing

## Abstract

The aim of this study was to examine the trend of average frequency of toothbrushing per day according to the sociodemographic characteristics using the 5th, 6th, and 7th Korean National Health and Nutrition Examination Survey (KNHANES) data for 9 years; from 2010 to 2018. It intends to understand the state of toothbrushing practices in South Korea, and to provide basic data for promoting toothbrushing practices. Data from the 5th, 6th, and 7th KNHANESs conducted by the Korean Disease Control and Prevention Agency each year was analyzed using IBM SPSS version 26.0 (IBM Co., Armonk, NY, USA). Tableau version 2020.1 was used for the graphs and geographic information system (GIS). The significance level of α was set at 0.05 for testing. In all analyses, the complex sampling analysis method with stratification variables, cluster variables, and weights was applied, and the complex sample linear regression analysis. The average frequency of toothbrushing per day was higher in women for all the years; these women belonged to the age group under 65 and the employed group. The average frequency of toothbrushing per day was lower in the married, elementary school graduate, part-time job, and low-income groups for all years. The demographic and social factors affecting the number of toothbrushing practices per day were gender, education level, income level, and work type. After analyzing the average frequency of toothbrushing per day in each of the cities and provinces, Gangwon-do and Jeonbuk showed the highest increase in the frequency of toothbrushing in 2018 compared to 2010, whereas Incheon showed the lowest increase. From the above results, the average frequency of toothbrushing per day was lower in males, those with a low education level, the unemployed, and those in the rural area for a period of 9 years. Therefore, in-depth oral health promotion projects and national benefit policies should be considered for effective toothbrushing education by identifying individuals who do not brush their teeth.

## 1. Introduction

In South Korea, the government and private organizations have been carrying out projects for oral disease prevention and early diagnosis. However, more than 85% of adults still have gingivitis and periodontitis, with poor oral health or accompanied by bad breath [1]. Since oral diseases are accompanied by bad breath due to poor oral hygiene, resulting in social and mental impairment, it is of utmost importance to prevent oral diseases from progressing [1]. Periodontal disease should be prevented rather than treated as it induces inflammation by bacteria toxins in the dental plaque adjacent to periodontal tissue and ultimately destroys the surrounding tissue. Therefore, it is possible to prevent oral disease by removing dental plaques, which is the cause of oral diseases. Dental plaques cannot be completely removed by physiological exercises such as the tongue or cheek movement and gargling. It is effectively removed by a physical method [2]. Toothbrushing is the basic and practical self-management method for dental plaques [3]. Toothbrushing removes food residues and stains, as well as dental plaques attached to the tooth surface. It is the basic and effective method to prevent oral diseases as it promotes keratinization of the epithelium by massaging the gums with a brush head to help blood circulation, and enhances resistance to infection [4]. Toothbrushing is effective only when the appropriate brushing method is used, but it is often impossible to completely remove dental plaques due to the insufficient brushing skills and the ignorance of the importance of toothbrushing [5]. The WHO emphasizes brushing twice a day and improving eating habits [6]. The study by Lee et al. [7] reported that 62.5% of the participants were brushing their teeth twice per day. On the other hand, the study by Choi et al. [2] reported that more than half of the participants (52.5%) brushed their teeth three times or more per day, but it was still insufficient as toothbrushing is recommended after each meal and before going to sleep. Therefore, in recognition of the importance of toothbrushing, South Korea has been making a lot of effort to develop the habit of correct toothbrushing in the people to improve oral health from the elementary school age. Elementary school teachers are being educated on the need for continuous oral health education and also encouraged to provide education on toothbrushing to the students [8]. However, education on toothbrushing is not provided for adults because it is done just once, which is inadequate. Since most education on toothbrushing for an adult group is conducted by providing information, it is limited in that, the effectiveness of inducing a change in toothbrushing habits cannot be realized [9]. Education on toothbrushing received at a dental hospital from a professional may result in no feedback on correct toothbrushing as it is provided only once in a lifetime. Since most adults have oral diseases, practical toothbrushing programs that can prove the effectiveness of oral health should be developed and applied. In addition, dental hospitals need to develop regular and systematic toothbrushing programs, and the policy-level support for regular toothbrushing education should be provided for effective oral care at low cost. Accordingly, we hypothesized that the average number of toothbrushing practices per day will vary depending on the sociodemographic characteristics of Koreans. This study aimed to confirm the change in the average frequency of toothbrushing per day in South Korean adults from 2010 to 2018, and to provide basic data for practical education by including adult oral health education in oral health policies. Until now, most of the studies related to toothbrushing have been conducted as short-term research, and there has been no study investigating the long-term trend of the average frequency of toothbrushing per day in South Korea.

This study examined the average frequency of toothbrushing per day according to sociodemographic characteristics for 9 years from 2010 to 2018 based on the 5th, 6th, and 7th Korean National Health and Nutrition Examination Survey (KNHANES) data. It intends to identify the state of toothbrushing practices in South Korea, and to provide basic data for promoting toothbrushing practices.

## 2. Materials and Methods

### 2.1. Participants

This study was conducted using data from the 5th, 6th, and 7th KNHANESs conducted by the Korean Disease Control and Prevention Agency each year. A total of 67,940 people—7542 in 2010; 7252 in 2011; 6801 in 2012; 6815 in 2013; 6420 in 2014; 6227 in 2015; 13,406 in 2016; 6751 in 2017; and 6726 in 2018—were selected as the study participants. KNHANES provides government-designated statistics in accordance with Article 17 of the Statistics Act (Approval No. 117002), and it has been conducted without the approval of the Institutional Review Board (IRB) after the first-year (2010-02CON-21-C), second-year (2011-02CON-06-C), and third-year (2012-01EXP-01-2C) surveys of the 5th KNHANES, and the first-year (2013-07CON-03-4C), second-year (2013-12EXP-03-5C), and third-year surveys of the 6th KNHANES, as it falls under the studies directly conducted for public welfare by the country pursuant to Article 2.1 of the Bioethics and Safety Act of its enforcement rule. From 2018, it was conducted with an IRB approval considering the collection of human materials and provision of raw data to third parties (Third-year Survey of 7th KNHANES Approval No. 2018-01-03-P-A).

### 2.2. Sociodemographic Characteristics

The participants were classified by sex and age groups. In South Korea, people aged 65 or older are classified as elderly, and based on this, the age group is divided into two groups.

In terms of marital status, the participants were classified into the married and unmarried groups. The married group was subdivided into: currently married, separated, widows/widowers, and the divorced for analysis. In terms of the level of education, the participants were classified into four groups: the elementary school, middle school, high school, and college graduate groups. In terms of the income level, the participants were classified into five groups: the lower, low-middle, middle, high-middle, and higher income groups. The participants were also classified into the employed and unemployed groups. Employed means a person who has been working for an hour or more for the purpose of income during the last week, and unemployed means a person who has not been working. The employed group was further classified by work type, into: the part-time, temporary, and permanent position groups. The residential areas used were classified into 16 cities and provinces, excluding Jeju-do, until 2015. Since 2016, it was classified into 17 cities and provinces including Jeju-do. The type of residential areas was classified into urban and rural areas. The criteria for cities and rural areas were divided based on the South Korean address cities and Dongeup-myeon.

### 2.3. Frequency of Toothbrushing per Day

The brushing time per day was marked as “0 for not brushing” and “1 for brushing” depending on whether or not the participants brushed their teeth before and after breakfast, before and after lunch, before and after dinner, after snacks, and before going to bed. Therefore, the frequency of toothbrushing per day was reported from the minimum of 0 to the maximum of 8 times a day.

### 2.4. Statistical Analysis

IBM SPSS version 26.0 (IBM Co., Armonk, NY, USA) was used for data analysis, and Tableau version 2020.1 was used for the graphs and geographic information system (GIS). The significance level α was set to 0.05 for testing. In all analyses, the complex sampling analysis method with stratification variables, cluster variables, and weights was applied, and the complex sample linear regression analysis of toothbrushing per day was compared according to the sociodemographic characteristics. All of the “unknown,” “not applicable,” and “missing” values of 8, 9, 88, and 99 were excluded, and unweighted frequencies were presented for the number of participants in all tables.

## 3. Results

### 3.1. Average Frequency of Toothbrushing per Day by Sex and Age

Comparing the average frequency of toothbrushing per day for 9 years, it increased each year. Excluding the 2015 year, there were significant differences (Figure 1), with a higher frequency in women than in men for all the years. The average frequency of toothbrushing per day by age was higher in the group under 65 than in the group aged 65 or older. There were significant differences in both sex and age (Figure 2). 

### 3.2. Average Frequency of Toothbrushing per Day by Marital Status

Comparing the average frequency of toothbrushing per day for 9 years, the average frequency of toothbrushing per day was higher in the unmarried group than in the married group from 2010 to 2013, whereas it was higher in the married group than in the unmarried group from 2014 to 2018. In addition, in terms of marital status, the spouse is dead status was the lowest on average, and there was a significant difference except for separated from spouse status (Figure 3).

### 3.3. Average Frequency of Toothbrushing per Day by Educational Level

Comparing the average frequency of toothbrushing per day for 9 years, the average frequency of toothbrushing per day was lowest in the elementary school graduate group for all the years, followed by the middle school, high school, and college graduate groups. There were significant differences in all groups (Figure 4).

### 3.4. Average Frequency of Toothbrushing per Day by Income Level

Comparing the average frequency of toothbrushing per day for 9 years, the average frequency of toothbrushing per day was lowest in the lower-income group for all years, followed by the low-middle, middle, high-middle, and higher groups. There were significant differences in all groups (Figure 5).

### 3.5. Average Frequency of Toothbrushing per Day by Employment Status and Work Type

Comparing the average frequency of toothbrushing per day for 9 years, the average frequency of toothbrushing per day was higher in the employed group than in the unemployed group for all the years, except for 2010. The average frequency of toothbrushing per day by work type was lowest in the participants of the part-time job group, followed by the temporary position, and permanent position groups. There were significant differences in all groups (Figure 6).

### 3.6. Average Frequency of Toothbrushing per Day by Type of Residential Area (Urban vs. Rural Areas)

Comparing the average frequency of toothbrushing per day for 9 years, the average frequency of toothbrushing per day was higher in the participants in urban areas than those in the rural areas for all the years. There were significant differences in all groups (Figure 7).

### 3.7. Average Frequency of Toothbrushing per Day by Residential Area (Cities and Provinces)

Comparing the average frequency of toothbrushing per day in the cities and provinces for 9 years, the average frequency of toothbrushing per day was highest in Incheon, and lowest in Gangwon-do in 2010. From 2011 to 2013, it was highest in Daejeon, and lowest in Gyeongnam. In 2014, it was highest in Gyeongbuk, and lowest in Gyeongnam. In 2015, it was highest in Gyeonggi-do, and lowest in Gangwon-do. In 2016, it was highest in Ulsan, and lowest in Gyeongbuk. In 2017, it was highest in Jeonbuk, and lowest in Gyeongbuk. In 2018, it was highest in Daejeon, and lowest in Jeju-do (Figure 8). The map was colored differently for each year, and within the same year, a darker shade of the color indicates a higher average frequency of toothbrushing per day.

## 4. Discussion

Since teeth play an essential role in mastication, the World Health Organization (WHO) declared that maintaining oral health is an essential factor for maintaining health. Oral health is determined by how well oral health behaviors are performed, and especially, the frequency of toothbrushing per day that is widely used as an index to predict healthy behavior [10]. The frequency of toothbrushing per day is a major variable representing the oral health behavior of an individual, which is not only a powerful factor influencing oral diseases, but is also closely associated with health risk behaviors [10,11]. According to a recent study conducted by Lee and Song [12], the risk of diabetes was increased in the group with periodontal disease and the group with high tooth loss, whereas the risk of diabetes was decreased in the group that received regular teeth scaling. Periodontal disease is also known to increase cerebrovascular disease, atherosclerosis, and premature birth [13,14]. Above all, the study reported that brushing teeth three or more times per day reduced the risk of diabetes by 8%, highlighting the importance of toothbrushing. WHO recommends brushing one’s teeth at least twice daily [6].

Comparing the average frequency of toothbrushing per day for 9 years since 2010, it increased progressively each year, but was still 2.66 times per day, not exceeding an average of 3 times per day. Brushing teeth three or more times a day not only reduces the risk of oral diseases but also the risk of cardiovascular diseases. Brushing teeth once more a day reduces the risk of cardiovascular diseases by about 9% [15], hence the need to increase the average frequency of toothbrushing per day to at least 3 times in South Korea. Moreover, it would be necessary to consider the sociodemographic characteristics when designing the toothbrushing promotion program. In this study, the average frequency of toothbrushing per day for 9 years was higher in women than in men for all the years, and it was higher in the group under 65 than in the group aged 65 or older. These results are similar with the results of the previous studies in which the frequency of toothbrushing decreased with age [16,17]. The average frequency of toothbrushing per day based on marital status was higher in the unmarried group than in the married group until 2013, whereas it was higher in the married group than in the unmarried group after 2014. The average frequency of toothbrushing per day tended to be higher in the high-education level and high-income groups. The average frequency of toothbrushing per day by employment status was higher in the employed group than in the unemployed group, and was lowest in the part-time job group, followed by the temporary position, and permanent position groups, respectively. The results are similar to those obtained by Shin et al. [18,19], who reported that the frequency of toothbrushing increased with an increase in socioeconomic status. Identifying the characteristics of the participants who are not likely to practice toothbrushing in preparation of active policy-level support would have a positive effect of promoting the habit of toothbrushing. Specifically, it is considered effective if the toothbrush education program is actively implemented in institutions where men with low social status and income work. The average frequency of toothbrushing per day was higher in urban areas than in rural area, and the average frequency of toothbrushing per day was highest in Incheon and lowest in Gangwon-do in 2010. In 2018, it was highest in Daejeon, and lowest in Jeju-do. Gangwon-do and Jeonbuk showed the highest increase in the frequency of toothbrushing in 2018 compared to 2010, whereas Incheon showed the lowest increase, with an insufficient increase in the average frequency of toothbrushing per day. Compared to the average frequency of toothbrushing per day in this study, while the area with the highest toothbrushing rate was the same, Daejeon, the area with the lowest frequency of toothbrushing per day was Jeju-do, which was different from the area of those brushing their teeth after lunch. The increase in toothbrushing after lunch was highest in Gyeongbuk, which was different from the results obtained in this study. Identifying the average frequency of toothbrushing per day in considering the geographic trends and the regional differences would provide useful data for preparing regional oral health promotion policies.

The toothbrushing education project in South Korea has a low level of awareness compared to other oral disease prevention projects [20]. However, toothbrushing is the most important and basic method of preventing oral diseases, and it would be necessary to expand and implement toothbrushing education as a continuous program, rather than a one-time class considering the characteristics of the participants. Toothbrushing education should be monitored and repeated training should be provided to increase the average frequency of toothbrushing per day. In addition, dental plaque management education, which is non-refundable by the health insurance as of now, should be designated as a refundable item to enable more South Koreans to have access to the education on toothbrushing by professionals without economic burden. It would be desirable not to limit the frequency of education to once in a lifetime, and to provide the dental plaque management education after each of the refundable scaling sessions provided once a year. This study has a limitation in that it is difficult to generalize the results as the analysis of this study has been conducted only based on a health examination survey. Since the data used in this study contained the results collected by objective interpretation in terms of the frequency of toothbrushing, a more in-depth analysis using subjective data would be required. Furthermore, we have not been able to check the annual trend according to the number of toothbrushing practices per day, so we will proceed with further research. Therefore, further research should be conducted by integrating the oral examination results and health examination questionnaire to examine not only the frequency of toothbrushing and oral health status, but also the association with systemic diseases.

## 5. Conclusions

This study aimed to examine the trend of average frequency of toothbrushing per day according to the sociodemographic characteristics based on the 5th, 6th, and 7th KNHANES data.

The average frequency of toothbrushing per day was higher in women than men for all the years, and it was higher in the group under 65 than in the group aged 65 or older.The average frequency of toothbrushing per day was lowest in those currently married for all the years, followed by the separated, the widows/widowers, and the divorced groups.The average frequency of toothbrushing per day was lowest in the elementary school graduate group for all the years, followed by the middle school, high school, and college graduate groups. The average frequency of toothbrushing per day by income level was lowest in the lower-income group, followed by the low-middle, middle, high-middle, and higher groups, respectively.The average frequency of toothbrushing per day was higher in the employed group than in the unemployed group for all the years, except for 2010. As for the average frequency of toothbrushing per day by work type, it was lowest in the part-time job group, followed by the temporary position and permanent position groups.After analyzing the average frequency of toothbrushing per day in each of the cities and provinces, Gangwon-do and Jeonbuk showed the highest increase in the frequency of toothbrushing in 2018 compared to 2010, whereas Incheon showed the lowest increase.

According to the above results, the average frequency of toothbrushing per day was lower in males, those with a low educational level, the unemployed, and in those in the rural area resident groups for 9 years. Therefore, in-depth oral health promotion projects and national benefit policies should be considered for effective toothbrushing education by identifying the participants with lack of toothbrushing according to sociodemographic characteristics.

## Figures and Tables

**Figure 1 ijerph-18-03522-f001:**
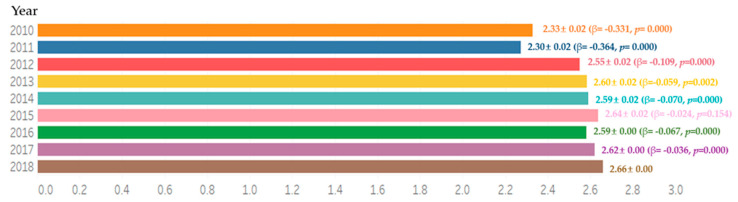
Average frequency of toothbrushing per day for 9 years. By complex sample linear regression analysis, M ± SD (β,*p*), R^2^ = 0.020, *p* = 0.000, reference category is 2018.

**Figure 2 ijerph-18-03522-f002:**
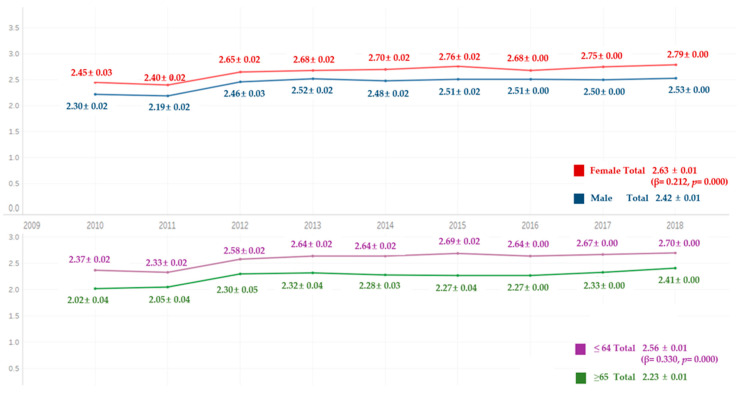
Average frequency of toothbrushing per day by gender and age. By complex sample linear regression analysis, M ± SD (β,*p*); R^2^ = 0.013, *p* = 0.000, reference category is male, R^2^ = 0.014, *p* = 0.000, reference category is 65 years or older.

**Figure 3 ijerph-18-03522-f003:**
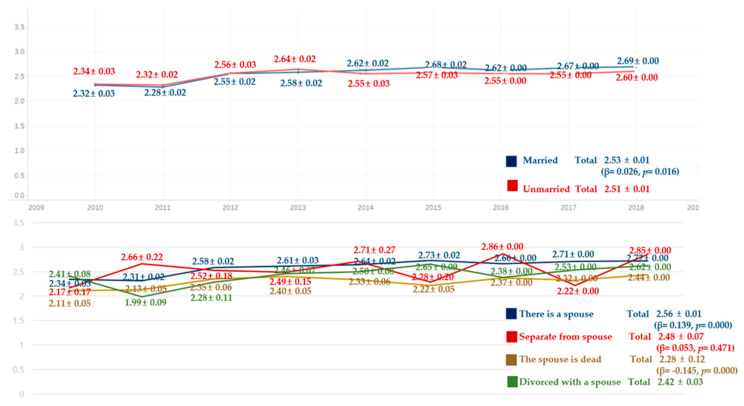
Average frequency of toothbrushing per day by marriage. By complex sample linear regression analysis, M ± SD (β,*p*), R^2^ = 0.000, *p* = 0.016, reference category is unmarried, R^2^ = 0.007, *p* = 0.000, reference category is divorced with a spouse.

**Figure 4 ijerph-18-03522-f004:**
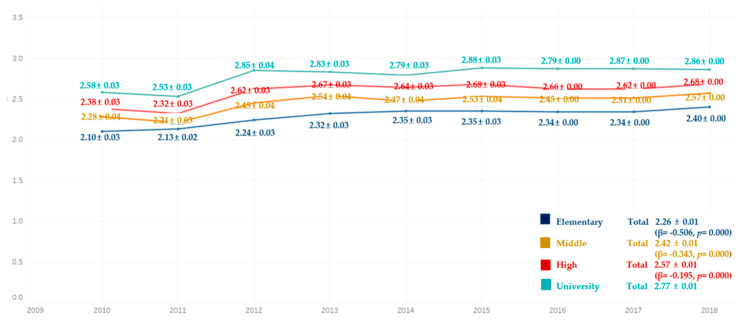
Average frequency of tooth-brushing per day by education level. By complex sample linear regression analysis, M ± SD (β,*p*), R^2^ = 0.046, *p* = 0.000, reference category is university.

**Figure 5 ijerph-18-03522-f005:**
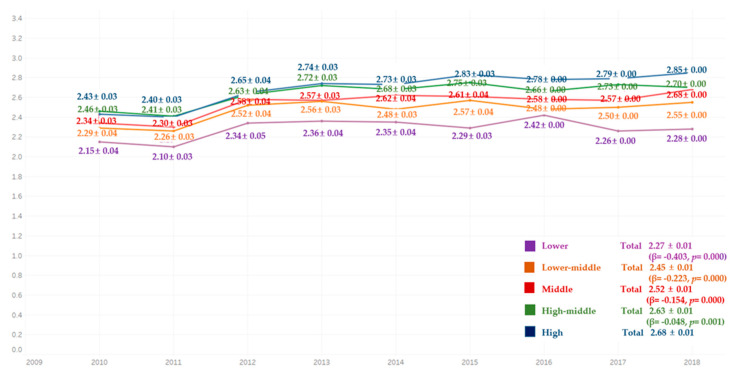
Average frequency of toothbrushing per day by income level. By complex sample linear regression analysis, M ± SD (β,*p*), R^2^ = 0.022, *p* = 0.000, reference category is high.

**Figure 6 ijerph-18-03522-f006:**
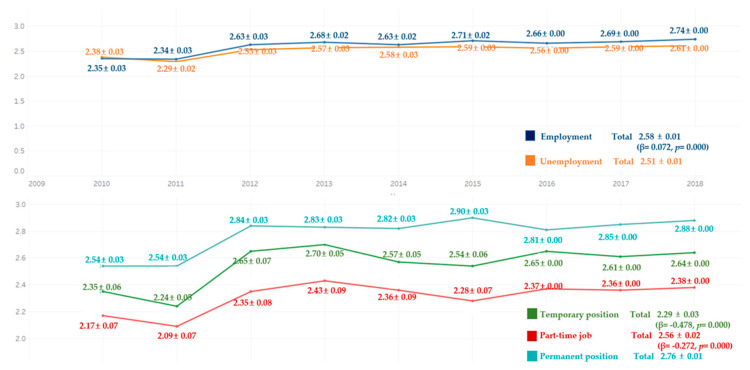
Average frequency of toothbrushing per day by work type. By complex sample linear regression analysis. R^2^ = 0.001, *p* = 0.000, reference category is unemployment, M ± SD (β,*p*), R^2^ = 0.025, *p* = 0.000, reference category is permanent position.

**Figure 7 ijerph-18-03522-f007:**
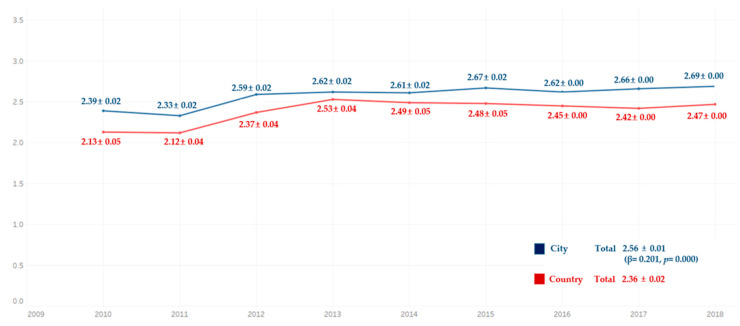
Average frequency of toothbrushing per day by residential area. By complex sample linear regression analysis, M ± SD (β,*p*), R^2^ = 0.007, *p* = 0.000, reference category is country.

**Figure 8 ijerph-18-03522-f008:**
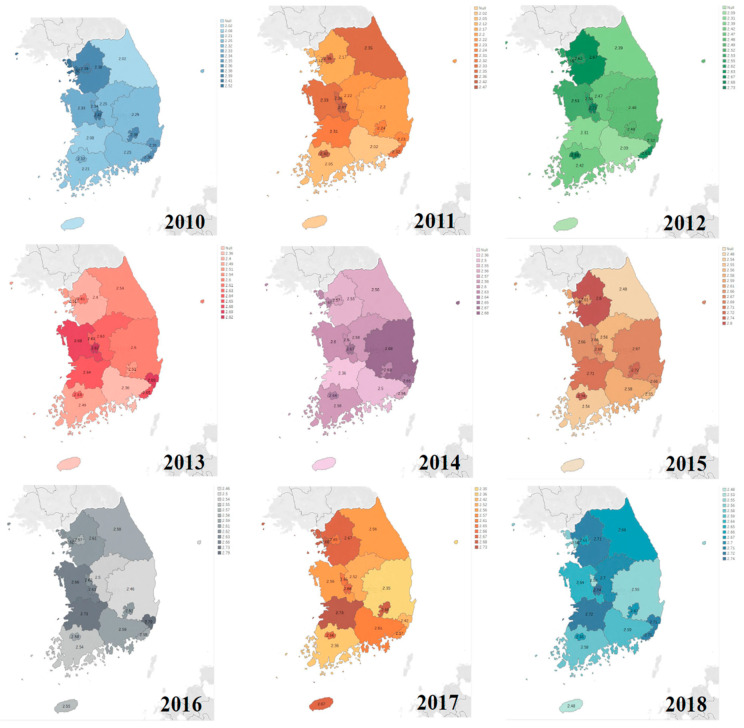
Average frequency of toothbrushing per day by area.

## Data Availability

The data presented in this study are available on request from the corresponding author. The data are not publicly available due to the restrictions of the social media platform.

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
