# Peer review of "Trend Analysis of Average Frequency Using Toothbrushing per Day in South Korea: An Observational Study of the 2010 to 2018 KNHANES Data"

_ijerph, 2021, doi:10.3390/ijerph18073522_

Round 1

Reviewer 1 Report

The article is interesting and well written. The methodology is clear and the results are correctly reported.

The benefits associated with toothbrushing are many, but the authors only mention diabetes and cardiovascular diseases - I think it would be interesting to mention more benefits and relationships / associations with other diseases - also please justify these associations.

The discussion can still be improved. There are many references to the results of the studies and to the results reported by other authors, but little importance is given to their justification - I believe this is truly important for the reader to understand why some parameters were evaluated and why some patients should be considered as at risk.

Author Response

Rev1. Comments and Suggestions for Authors

The article is interesting and well written. The methodology is clear and the results are correctly reported.

The benefits associated with toothbrushing are many, but the authors only mention diabetes and cardiovascular diseases - I think it would be interesting to mention more benefits and relationships / associations with other diseases - also please justify these associations.

à We added the contents to the discussion.

The discussion can still be improved. There are many references to the results of the studies and to the results reported by other authors, but little importance is given to their justification - I believe this is truly important for the reader to understand why some parameters were evaluated and why some patients should be considered as at risk.

à We added the contents to the discussion.

Reviewer 2 Report

The manuscript submitted to IJERPH entitled “Analysis of the Trend of Average Frequency of Toothbrushing per Day According to Sociodemographic Characteristics in South Korea: using the 2010 to 2018 KNHANES Data” is an original article which aim to evaluate the trend of average frequency of toothbrushing per day according to the sociodemographic characteristics using the 5th, 6th, and 7th Korean National Health and Nutrition Examination Survey.

On my opinion the article is interesting, well written, with good English.

However, I highlighted some minor issues.

  • Title: Please improve, it is no clear.
  • Abstract: Please structure the abstract to attract the reader's attention,and to adapt it accordingly.
  • Introduction: This section has been properly prepared.
  • Materials and methods: This section has been properly prepared.
  • Results: This section has been properly prepared.
  • Discussion: This section has been properly prepared.
  • Conclusion: This section has been properly prepared.
  • Figures: Please improve figures quality
  • English language: Minor spell check is required.

After making the indicated changes, the article may be suitable for publication.

Author Response

Rev2. Comments and Suggestions for Authors

Title: Please improve, it is no clear.

->We revised the contents.

Abstract: Please structure the abstract to attract the reader's attention and to adapt it accordingly.

->We revised the contents.

Introduction: This section has been properly prepared.

Materials and methods: This section has been properly prepared.

Results: This section has been properly prepared.

Discussion: This section has been properly prepared.

Conclusion: This section has been properly prepared.

Figures: Please improve figures quality.

-> We revised the contents.

English language: Minor spell check is required.

-> We were checked English by an expert.

After making the indicated changes, the article may be suitable for publication.

Reviewer 3 Report

The paper shows the change in toothbrushing frequency among Korean people using national database in a cross-sectional study.

This is an interesting study. However, the paper has issues. The paper needs to be revised.

First, the logic is unclear. Second, there are no multivariate analyses. Third, the authors should follow the STROBE guideline.

TITLE

  • The title should include a design following the STROBE guideline.

ABSTRACT

  • The abstract should be fully changed after the revision of main text.

INTRODUCTION

  • The logic is unclear. The aim did not match with the conclusion. Furthermore, there is no hypothesis. The main outcome should be changed from toothbrushing frequency to periodontal disease, because the authors mentioned the relationship in the introduction and the outcome is better for public health. The current form is not manuscript and just a report, because there is no appropriate logic. What are new findings in this study? What do the authors want to add for public health? For example, a PECO model might be recommended; P, Korean people based on the national data; E, high frequency of toothbrushing, C; low frequency of toothbrushing, O; prevalence of periodontal disease.
  • Please add a hypothesis before aim.
  • Please add a reference (“more than 85%”, L33).
  • Please refer original papers. The reference 4 is true?
  • The reference 6 recommends three-time toothbrushing. Is it correct? Furthermore, the “three times” is not consensus among periodontology and nobody knows the clinical relevance.

MATERIALS AND METHODS

  • Please revise the whole parts following the STROBE checklist. There are many lack parts including the eligibility criteria, variables, bias, study size, quantitative variables, statistical methods.
  • Why did the authors use the cut-off value, 65 years old? Please add references and reason in the text.
  • How did the authors define rural and urban areas? Please add the detail definitions using appropriate references.
  • The authors used only toothbrushing frequency in the national survey. Please add more data and analyze them deeply based on a new logic. The current form is not acceptable.
  • Please add some multivariate analyses to seek the interaction and/or adjust the confounders.

RESULTS

  • Please revise the results following the new design.
  • Figures had no variations like SD and no statistical analyses.

DISCUSSION

  • Please revise the discussion and conclusion following the new results.
  • Please add more comments about limitation, such as no data of important confounders, no connected data (each person in each year), etc.

Author Response

Rev3. Comments and Suggestions for Authors

First, the logic is unclear. Second, there are no multivariate analyses. Third, the authors should follow the STROBE guideline.

TITLE

The title should include a design following the STROBE guideline.

à We revised the contents.

ABSTRACT

The abstract should be fully changed after the revision of main text.

à We revised the contents.

INTRODUCTION

The logic is unclear. The aim did not match with the conclusion. Furthermore, there is no hypothesis. The main outcome should be changed from toothbrushing frequency to periodontal disease, because the authors mentioned the relationship in the introduction and the outcome is better for public health.

à The purpose of this study is to check the frequency of brushing according to demographic characteristics and does not include clinical diagnosis that can confirm periodontal disease. However, the reason why the introduction states that oral and systemic diseases are related to brushing is because we want to provide basic data so that we can brush our teeth well. In the next study, we will check the relevance of toothbrushing based on data with oral examination. Thank you.

The current form is not manuscript and just a report, because there is no appropriate logic. What are new findings in this study? What do the authors want to add for public health? For example, a PECO model might be recommended; P, Korean people based on the national data; E, high frequency of toothbrushing, C; low frequency of toothbrushing, O; prevalence of periodontal disease.

à The purpose of this study is to identify the average daily changes in toothbrushes depending on the demographic and sociological characteristics of Koreans over the past nine years. Therefore, we would like to provide basic data so that we can brush our teeth well by identifying the classes that lack the number of toothbrushing. However, according to the your opinion, we will plan a study to check the prevalence of periodontal disease according to the number of toothbrushes per day. Also, we suggested that we will conduct a study to check the connection with systemic diseases using oral examination data at the limit.

 Please add a reference (“more than 85%”, L33). Please refer original papers. The reference 4 is true? The reference 6 recommends three-time toothbrushing. Is it correct? Furthermore, the “three times” is not consensus among periodontology and nobody knows the clinical relevance.

à We checked the reference. The World Health Organization (WHO) recommends brushing your teeth at least twice a day, which is data from 2015 and is out of date. South Korea recommends three toothbrushes after meals, according to reports that the incidence of systemic diseases decreases as the number of toothbrushes increases per day. We checked the reference again and revised the contents.

MATERIALS AND METHODS

Please revise the whole parts following the STROBE checklist. There are many lack parts including the eligibility criteria, variables, bias, study size, quantitative variables, statistical methods.

à We revised according to the STROBE specification.

Why did the authors use the cut-off value, 65 years old? Please add references and reason in the text. How did the authors define rural and urban areas? Please add the detail definitions using appropriate references.

à We revised the contents.

The authors used only toothbrushing frequency in the national survey. Please add more data and analyze them deeply based on a new logic. The current form is not acceptable.

à What we want to check in this study is the number of toothbrushes per day by region. This is to

Please add some multivariate analyses to seek the interaction and/or adjust the confounders.

à We revised the contents.

RESULTS

Please revise the results following the new design.

à We revised the contents.

Figures had no variations like SD and no statistical analyses.

à We revised the contents.

DISCUSSION

Please revise the discussion and conclusion following the new results.

à We revised the contents.

Please add more comments about limitation, such as no data of important confounders, no connected data (each person in each year).

à The results of checking the average number of toothbrushes per day by inserting all variables of demographic and sociological characteristics are presented in Table 1.

Round 2

Reviewer 3 Report

The paper was overall improved. However, there are some issues. The authors did not answer all questions. Furthermore, there are typos, such as “Coutry” and “cities and provinces”. The paper needs to be revised.

1) Please add the design in the title; “an observational study”.

2) Please add the hypothesis before the aim; “We hypothesized that ///”.

3) In the materials and methods, please add the eligibility criteria; i.e., the inclusion criteria were…, and the exclusion criteria were ….

Please add some comments about bias and study size.

Please add a characteristic table and comments in the text.

The statistical analyses were not enough. The authors should add Tukey test to investigate the difference in the average frequency of toothbrushing per day between 2010, 2011, …2018.

4) Figures had no variations like SD (error bars) and some figures had no statistical analyses. Furthermore, the authors mentioned “By complex sample linear regression analysis,” in the figure legends. They cannot add the results of linear regression analysis and the words in the figures.

5) Please DO NOT combine all data to perform the multiple regression analysis. The authors should analyze the data in each year because the aim was to examine the trend of average frequency of toothbrushing per day. Furthermore, based on the aim, the authors should show the trend of 3 times per day, 2 times per day and other groups in new figure.

6) The conclusion is still inappropriate. The authors can only state that the average frequency of toothbrushing per day was significantly associated with … , and the frequency tended to be increased based on the 5th, 6th, and 7th KNHANES data.
